# The relationship between home modifications and frailty among older adults: a scoping review protocol

Julius Ernst[1], Elisabeth Zwinge[1,2], Maximilian König [1,2]*

1 Department of Internal Medicine D – Geriatrics, University Medicine Greifswald, Greifswald, Germany,
2 Housing and Digitalization Competence Center Mecklenburg-Vorpommern, Wolgast/Schwerin, Germany

* maximilian.koenig@med.uni-greifswald.de

The relationship between home modifications
and frailty among older adults: a scoping
review protocol. PLoS One 20(11): e0335822.

KOREA, REPUBLIC OF

**Peer Review History:** PLOS recognizes the
benefits of transparency in the peer review
process; therefore, we enable the publication
of all of the content of peer review and
author responses alongside final, published
articles. The editorial history of this article is
available here: https://doi.org/10.1371/journal.
pone.0335822

## Abstract

### Introduction

The rapid aging of populations worldwide presents significant challenges to health-care and social systems. Frailty—a multidimensional geriatric syndrome marked by physical weakness, exhausted reserve capacities across all organ systems, and increased susceptibility to adverse events—is prevalent among older adults. Ensuring an optimal fit between individuals and their living environment is vital for preserving independence and reducing the burden on both professional healthcare services and informal care. A supportive home environment may help delay the onset or progression of frailty, while a person-environment misfit is likely to accelerate physical decline. Home modifications may play a critical role in this regard. This scoping review seeks to identify, synthesize and map the current evidence on the associations between accessible housing, home modifications, and frailty in community-dwelling older adults aged 65 and above. The aim is to enhance understanding of these relationships, highlight evidence gaps, and support the development of suitable interventions.

### Methods and analysis

This scoping review will follow the framework developed by Arksey and O'Malley and will adhere to the guidelines provided by the Joanna Briggs Institute (JBI) and the PRISMA-ScR checklist. A systematic search will be conducted in PubMed, Embase, the Cochrane Library, Google Scholar, and CINAHL. Rayyan will facilitate deduplication and screening. Titles, abstracts, and full texts will be independently assessed by two reviewers against eligibility criteria. A structured data extraction form will be used to collect information on the study type, publication year, research design, sample characteristics, housing accessibility indicators, methods of frailty assessment, types of home modifications, and key findings.

**Data availability statement:** No datasets were generated or analysed during the current study. All relevant data from this study will be made available upon study completion.

**Funding:** The work is conducted within the scope of a publicly funded project, the Housing and Digitalisation Competence Center Mecklenburg-Vorpommern (Landesfachstelle für Wohn- und Digitalisierungsberatung), funded by the Ministry of Social Affairs, Health and Sport of Mecklenburg-Vorpommern. The authors MK and EZ serve as the project leads. JE declares no conflicts of interest. The funders had no role in study design, data collection and analysis, decision to publish, or preparation of the manuscript.

**Competing interests:** The work is conducted within the scope of a publicly funded project, the Housing and Digitalisation Competence Center Mecklenburg-Vorpommern (Landesfachstelle für Wohn- und Digitalisierungsberatung), funded by the Ministry of Social Affairs, Health and Sport of Mecklenburg-Vorpommern. The authors MK and EZ serve as the project leads. JE declares no conflicts of interest. The funders had no role in study design, data collection and analysis, decision to publish, or preparation of the manuscript.

## Dissemination

Findings will be disseminated in a peer-reviewed journal and presented at conferences.

---

## Background and rationale

Demographic trends indicate a rapidly aging global population, posing profound implications for healthcare and social support systems [1]. Aging increases the risk of age-related diseases and functional decline, notably the development of frailty [2]. Frailty, a multifactorial geriatric syndrome characterized by diminished physiological reserves and increased vulnerability to stressors, is strongly associated with adverse outcomes including falls, hospitalizations, care dependency, and mortality [2,3]. Importantly, the development of frailty is currently understood as a dynamic and potentially reversible process, shaped by intrinsic and extrinsic factors and can be positively influenced by targeted measures [4,5].

Although research on frailty has emphasized individual-level risk factors such as multimorbidity, inactivity, and nutritional deficits, the significance of environmental determinants is increasingly recognized [6–10]. Environmental gerontology explores person-environment interactions, positing that person–environment fit is critical to successful aging [11–14]; a misalignment between an individual's abilities and their living environment can exacerbate functional decline and dependency [9,15]. In addition to objective housing conditions, subjective aspects, such as emotional ties to the place of residence are important in the analysis and design of age-appropriate living spaces [16]. It has been shown that accessible layouts, as well as home modification interventions for older adults, encompassing both structural and functional adaptations to the home environment, such as grab bars, non-slip flooring, and appropriate lighting, can reduce barriers in the home environment, enhance safety and well-being, reduce fall risk, reduce functional difficulties, improve participation and perceptions of safety and exertion, prevent and delay care dependence, reduce care home admissions, as well as hospital admissions, and potentially delay the onset and progression of frailty [9,10,13,17–25]. For example, the evaluation of the Care & Repair Cymru program in Wales showed that such interventions have a protective effect on nursing home admissions, particularly in moderately to severely frail people [17]. In Japan, a retrospective cohort study demonstrated an association between home adaptations and delayed decline in care needs [25]. It also has been shown that home modifications reduce difficulties experienced by older people in everyday activities – and that this effect lasts up to six months after installation [19]. Long-term data also showed that every additional month of waiting for a housing adaptation is associated with more difficulties in dealing with everyday tasks, which underlines the importance of rapid implementation [19]. Others point to the importance of social and subjective elements, such as emotional attachment to one's home, perceived safety, and social interaction [6–8,26,27]. A unifying understanding of the complex interplay of home and health in older adults is still lacking [18,21,22,28]. Frailty in particular,

has so far been investigated less frequently as an outcome compared to other endpoints, such as falls, and has not yet been the focus of a systematic review. This is despite growing evidence that both objective and subjective aspects of the home environment are predictors of frailty in older people [6,20,29]. Although the link between housing accessibility, home adaptations, and frailty appears intuitive, the current evidence base is markedly heterogeneous, with significant variation in study designs, intervention approaches, and frailty assessment methods. It is also fragmented, and a comprehensive overview is lacking.

Therefore, a scoping review is well-suited to chart the literature, identify research gaps, and help shape future research directions and policy measures. This is especially important because both more observational and interventional research approaches, including randomized controlled trials, are needed in this area. Understanding the relationship between housing characteristics, home modifications and frailty is critical for designing effective programs that support aging in place and reduce the healthcare burden associated with frailty [27]. Given the growing challenge of caring for an increasing number of frail older adults with functional limitations amid a shrinking working-age population, efforts to prevent frailty and to support the autonomy and quality of life of older people have become a public health priority. In this context, the findings are expected to inform public health policies and guidelines on housing accessibility and environmental adaptations for older adults.

## Objectives

This scoping review aims to identify, describe, map, and synthesize existing research on the – presumably bidirectional – relationship between housing accessibility, home modifications and frailty among community-dwelling adults aged ≥65 years. Specifically, we will:

• Examine types of home modification interventions, study populations, outcome measures, and study designs.

• Explore evidence on the impact of accessible housing and home modifications on the onset, progression, or reversal of frailty.

The findings of this scoping review aim to provide a basis for future research and contribute to practical recommendations in the field of housing modification.

## Methods and analysis

### Design and framework

We will conduct this scoping review following Arksey and O'Malley's framework and guidance from the JBI (Joanna Briggs Institute) and the PRISMA-ScR checklist in S1 File [30–32]. The project has been preregistered on the Open Science Framework (OSF) to ensure transparency and reproducibility (osf.io/6xajk). Any amendments to the protocol will be documented there.

### Search strategy

An initial exploratory search in PubMed identified relevant studies on the topic, from which key terms and keywords were extracted. These informed the development of a comprehensive search strategy aimed at capturing the breadth of relevant literature (Table 1). Using this refined strategy, a second, systematic search will be conducted in PubMed, Embase, the Cochrane Library, and CINAHL. Additionally, the top 250 hits from Google Scholar will be screened for relevance to capture grey literature. References of included studies will be hand-searched to ensure completeness.

### Study selection

Rayyan [33] will be used to manage references, remove duplicates, and support blind screening by two independent reviewers. First, titles and abstracts will be screened against eligibility criteria. Next, full texts will be assessed, with

**Table 1. Preliminary version of the search strategy (PubMed).**

| Domain | Keywords |
|---|---|
| **Frailty** | frailty [TIAB] OR frail [TIAB] OR prefrail [TIAB] |
| **Older adults** | "old people" [TIAB] OR "older people" [TIAB] OR aged [TIAB] OR aging [TIAB] OR elderly [TIAB] OR "older adult*" [TIAB] OR "old population" [TIAB] OR senior* [TIAB] OR "elders" [TIAB] |
| **Home/ Environment/ Accessibility** | "home modification*" [TIAB] OR accessibility [TIAB] OR "home environment*" [TIAB] OR "housing environment*" [TIAB] OR "environmental barrier*" [TIAB] OR "housing modification*" [TIAB] OR "building environment*" [TIAB] OR "built environment*" [TIAB] OR "home advice*" [TIAB] OR "environmental intervention*" [TIAB] OR "modification intervention*" [TIAB] OR "assistive device*" [TIAB] OR "assistive technology" [TIAB] OR "environmental attribute*" [TIAB] OR "household environment*" [TIAB] OR "environmental factor*" [TIAB] OR "housing improvement*" [TIAB] OR "accessible housing*" [TIAB] OR "home assessment*" [TIAB] |

TIAB = Title/Abstract

disagreements resolved by discussion. Reasons for excluding studies after full-text review will be documented and reported in the final scoping review. The selection process will be documented in a PRISMA-ScR flowchart.

## Eligibility criteria

Studies will be included if they:

- include community-dwelling adults aged ≥65 (Population),

- examine home modifications and/or accessibility as exposures, interventions, or contextual factors (Concept), and

- explore frailty prevention, delay, or management as an outcome (Context),

    according to the PCC (Population, Concept, Context) framework [34]. Only English-language studies published up to September 2025 will be included.

## Data extraction

Data will be extracted independently by two reviewers using a piloted extraction form. Extracted information will include study design, setting, characteristics of the studied population, definitions and measures of frailty (including the assessment tools used), assessments of environmental accessibility or types of home modifications and interventions to improve accessibility, as well as the key findings. The data extraction form will initially be tested on 10 studies and, if necessary, adapted to ensure clarity and completeness in an iterative process. The methodological quality and potential risk of bias of the included studies will also be checked using the JBI criteria.

## Data synthesis

Extracted data will be collated in Excel. Tables and figures will be used to map the evidence, highlight key themes, and identify gaps in the literature. A narrative summary will then integrate and interpret the findings, providing an overview of patterns and relationships between housing accessibility, home modifications, and frailty among community-dwelling older adults.

## Ethics and Dissemination

No ethical approval is required as this review uses only published literature. Findings will be disseminated in peer-reviewed journals and presented at conferences.

## Patient and Public Involvement

Neither patients nor members of the public will be involved in the design or conduct of this review.

## Expected Limitations

This scoping review may face several limitations. We anticipate high heterogeneity among included studies with respect to study design, types of home/environmental assessments and housing modification interventions, and methods used to assess frailty, which may limit the comparability and synthesis of findings.

## Supporting information

**S1 File. PRISMA Checklist.**
(PDF)

## Author contributions

**Conceptualization:** Julius Ernst, Maximilian König.

**Formal analysis:** Elisabeth Zwinge, Maximilian König.

**Investigation:** Julius Ernst, Elisabeth Zwinge, Maximilian König.

**Methodology:** Julius Ernst, Elisabeth Zwinge, Maximilian König.

**Project administration:** Maximilian König.

**Resources:** Maximilian König.

**Supervision:** Maximilian König.

**Validation:** Elisabeth Zwinge.

**Writing – original draft:** Julius Ernst, Maximilian König.

**Writing – review & editing:** Julius Ernst, Elisabeth Zwinge, Maximilian König.

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
