## [Decision Letter · Decision Letter 0]

24 Sep 2025

Dear Dr. König,

Thank you for submitting your manuscript to PLOS ONE. After careful consideration, we feel that it has merit but does not fully meet PLOS ONE’s publication criteria as it currently stands. Therefore, we invite you to submit a revised version of the manuscript that addresses the points raised during the review process.

We look forward to receiving your revised manuscript.

Kind regards,

Mi-So Shim

Academic Editor

PLOS ONE

Journal Requirements:

“The author MK serves as the project lead of the Housing and Digitalisation Competence Center Mecklenburg-Vorpommern (Landesfachstelle für Wohn- und Digitalisierungsberatung), a publicly funded initiative. The project is funded by the Ministry of Social Affairs, Health and Sport of Mecklenburg-Vorpommern. No additional financial support from private or commercial entities was received in relation to this work.”

“I have read the journal's policy and the authors of this manuscript have the following competing interests: The authors declare no financial or non-financial competing interests. The work is conducted within the scope of a publicly funded state-level office supported by the federal state of Mecklenburg-Vorpommern.”

Reviewers' comments:

Reviewer's Responses to Questions

**Comments to the Author**

1. Does the manuscript provide a valid rationale for the proposed study, with clearly identified and justified research questions?

Reviewer #1: Yes

Reviewer #2: Yes

Reviewer #3: Yes

2. Is the protocol technically sound and planned in a manner that will lead to a meaningful outcome and allow testing the stated hypotheses?

Reviewer #1: Yes

Reviewer #2: Yes

Reviewer #3: Partly

3. Is the methodology feasible and described in sufficient detail to allow the work to be replicable?

Reviewer #1: Yes

Reviewer #2: Yes

Reviewer #3: Yes

4. Have the authors described where all data underlying the findings will be made available when the study is complete?

Reviewer #1: Yes

Reviewer #2: Yes

Reviewer #3: No

5. Is the manuscript presented in an intelligible fashion and written in standard English?

Reviewer #1: Yes

Reviewer #2: Yes

Reviewer #3: Yes

You may also provide optional suggestions and comments to authors that they might find helpful in planning their study.

Reviewer #1: The methodological proposal is solid, coherent, and aligned with the main conceptual frameworks and international guidelines for scoping reviews, such as the Arksey and O'Malley model, the Joanna Briggs Institute (JBI) guidelines, and the PRISMA-ScR checklist. The proposed methodology is adequate to meet the stated objectives and has high potential to generate relevant and useful evidence. The objectives are clearly formulated and focus on mapping and synthesizing the available evidence on the topic of interest. The application of the PCC (Population, Concept, Context) framework allows for an appropriate and relevant selection of studies. It is recommended that the protocol explicitly state the expected limitations, such as the possible heterogeneity among the included studies (in terms of design, intervention, and measurement of frailty). In addition to including the search time frame. It would be relevant to incorporate a section addressing the potential use of the findings for the design of home-based interventions and for the formulation of public policies.

Reviewer #2: The Article was presented in a comprehensive, yet concse manner, the aims and objectives well put forward, Above all- a Great scoping review.

Reviewer #3: Thank you for sending this article in its protocol version; here are some comments:

Methods:

Consider the protocol log.

It would be important to detail the search strategy and/or keywords to use.

Additionally, it is important to define which type of test will be used to measure frailty, whether home modifications or accessibility will be evaluated in relation to frailty prevention and management outcomes, and how the frailty variable will be grouped or categorised for analysis.

It is important to include PRISMA-P as supplementary material because, although it was designed for systematic reviews, it is a protocol checklist that could be used for scoping reviews.

**Do you want your identity to be public for this peer review?** For information about this choice, including consent withdrawal, please see our Privacy Policy

Reviewer #1: No

Reviewer #2: No

Reviewer #3: No

---

## [Author Response · Author response to Decision Letter 1]

6 Oct 2025

Response to Reviewers (point-by-point):

Response: Checked

“The author MK serves as the project lead of the Housing and Digitalisation Competence Center Mecklenburg-Vorpommern (Landesfachstelle für Wohn- und Digitalisierungsberatung), a publicly funded initiative. The project is funded by the Ministry of Social Affairs, Health and Sport of Mecklenburg-Vorpommern. No additional financial support from private or commercial entities was received in relation to this work.”

Response: We added: "The funders had no role in study design, data collection and analysis, decision to publish, or preparation of the manuscript."

We included this statement in our revised cover letter and in the manuscript.

Response: We added the statement: "This does not alter our adherence to PLOS ONE policies on sharing data and materials.” in our updated Competing Interests statement in our cover letter.

Response: We are committed to adhering to the open data policy. As this is a protocol, no primary data have been collected yet, so there are no datasets to share at this stage.

Response: not applicable

Response: checked

Reviewers' comments:

Reviewer's Responses to Questions

Comments to the Author

Reviewer #1: The methodological proposal is solid, coherent, and aligned with the main conceptual frameworks and international guidelines for scoping reviews, such as the Arksey and O'Malley model, the Joanna Briggs Institute (JBI) guidelines, and the PRISMA-ScR checklist. The proposed methodology is adequate to meet the stated objectives and has high potential to generate relevant and useful evidence. The objectives are clearly formulated and focus on mapping and synthesizing the available evidence on the topic of interest. The application of the PCC (Population, Concept, Context) framework allows for an appropriate and relevant selection of studies. It is recommended that the protocol explicitly state the expected limitations, such as the possible heterogeneity among the included studies (in terms of design, intervention, and measurement of frailty). In addition to including the search time frame. It would be relevant to incorporate a section addressing the potential use of the findings for the design of home-based interventions and for the formulation of public policies.

Response: We appreciate the reviewer’s thoughtful suggestions.

Ad Limitations: We have revised the protocol to explicitly outline the expected limitations of our review, particularly the potential heterogeneity among included studies. We have added a new “Limitations” section to the protocol.

Ad Search Time Frame: We have now clearly stated the time frame for the literature search in the “Search Strategy” section. The review will include studies published up to September 2025.

Ad Implications for Practice and Policy: we expanded the part on this in the “Background and Rationale” section.

Reviewer #2: The Article was presented in a comprehensive, yet concise manner, the aims and objectives well put forward, Above all- a Great scoping review.

Response: Thank you.

Reviewer #3: Thank you for sending this article in its protocol version; here are some comments:

Methods:

Consider the protocol log.

Response: We appreciate this recommendation. We have pre-registered this project on Open Science Framework (OSF) [osf.io/6xajk]. Any amendments to the protocol will be documented there (= protocol log) and reported transparently in the final publication.

It would be important to detail the search strategy and/or keywords to use.

Response: We appreciate this recommendation. We have now included a more detailed (preliminary) search strategy in the Methods section of the protocol, including example keywords and Boolean operators.

Additionally, it is important to define which type of test will be used to measure frailty, whether home modifications or accessibility will be evaluated in relation to frailty prevention and management outcomes, and how the frailty variable will be grouped or categorised for analysis.

Response: Thank you for this feedback. As this is a scoping review, we do not aim to restrict the included studies to specific frailty assessment tools. However, we have clarified in the protocol that we will extract and report the type of frailty assessment used in each study.

We have also clarified that we will chart whether the study focuses on assessments of environmental accessibility home modifications, accessibility interventions, or both, and how these are linked to frailty prevention, mitigation, or management outcomes.

It is important to include PRISMA-P as supplementary material because, although it was designed for systematic reviews, it is a protocol checklist that could be used for scoping reviews.

Response:

Thank you for this suggestion. We have completed the PRISMA-ScR (Preferred Reporting Items for Systematic reviews and Meta-Analyses extension for Scoping Reviews) Checklist checklist and included it as a Supplementary File.

---

## [Editor Report · Decision Letter 1]

17 Oct 2025

The Relationship Between Home Modifications and Frailty Among Older Adults: A Scoping Review Protocol

PONE-D-25-36496R1

Dear Dr. König,

We’re pleased to inform you that your manuscript has been judged scientifically suitable for publication and will be formally accepted for publication once it meets all outstanding technical requirements.

Kind regards,

Mi-So Shim

Academic Editor

PLOS ONE

---

## [Editor Report · Acceptance letter]

PONE-D-25-36496R1

PLOS ONE

Dear Dr. König,

I'm pleased to inform you that your manuscript has been deemed suitable for publication in PLOS ONE. Congratulations! Your manuscript is now being handed over to our production team.

Kind regards,

on behalf of

Dr. Mi-So Shim

Academic Editor

PLOS ONE